# Anti-HCV antibody titer highly predicts HCV viremia in patients with hepatitis B virus dual-infection

**Hung-Yin Liu[1][◦], Yi-Hung Lin[1][◦], Pei-Ju Lin[2], Pei-Chien Tsai[1], Shu-Fen Liu[1], Ying-Chou Huang[1], Jia-Jiun Tsai[1], Ching-I Huang[1,3], Ming-Lun Yeh[1,3], Po-Cheng Liang[1], Zu-Yau Lin[1,2], Chia-Yen Dai[1,3], Jee-Fu Huang[1,3], Wan-Long Chuang[1,3], Chung-Feng Huang[1,3]\*, Ming-Lung Yu[1,3,4,5]**

**1** Hepatobiliary Division, Department of Internal Medicine, Kaohsiung Medical University Hospital, Kaohsiung Medical University, Kaohsiung, Taiwan, **2** Department of Nursing, Kaohsiung Medical University Hospital, Kaohsiung Medical University, Kaohsiung, Taiwan, **3** Faculty of Internal Medicine, School of Medicine, College of Medicine, Kaohsiung Medical University, Kaohsiung, Taiwan, **4** Institute of Biomedical Sciences, National Sun Yat-Sen University, Kaohsiung, Taiwan, **5** College of Biological Science and Technology, National Chiao Tung University, Hsin-Chu, Taiwan

◦ These authors contributed equally to this work.
\* fengcheerup@gmail.com

**Data Availability Statement:** All relevant data are within the paper and its Supporting Information files.

## Abstract

### Background/Aims

Hepatitis C Virus (HCV) infection is diagnosed by the presence of antibody to HCV and/or HCV RNA. This study aimed to evaluate the accuracy of anti-HCV titer (S/CO ratio) in predicting HCV viremia in patients with or without hepatitis B virus (HBV) dual infection.

### Methods

Anti-HCV seropositive patients who were treatment-naïve consecutively enrolled. Anti-HCV antibodies were detected using a commercially chemiluminescent microparticle immunoassay. HCV RNA was detected by real-time PCR method.

### Results

A total of 1321 including1196 mono-infected and 125 HBV dually infected patients were analyzed. The best cut-off value of anti-HCV titer in predicting HCV viremia was 9.95 (AUROC 0.99, P<0.0001). Of the entire cohort, the anti-HCV cut-off value of 10 provided the best accuracy, 96.8%, with the sensitivity, specificity, positive predictive value (PPV) and negative predictive value (NPV) of 96.3%, 98.9%, 99.7% and 87.3% respectively. The best cut-off value of anti-HCV titer in predicting HCV viremia was 9.95 (AUROC 0.99, P<0.0001) and 9.36 (AUROC 1.00, P<0.0001) in patients with HCV mono-infection and HBV dual-infection respectively. Among the HBV dually infected patients, the accuracy of anti-HCV titer in predicting HCV viremia reached up to 100% with the cut-off value of 9. All the patients were HCV-viremic if their anti-HCV titer was greater than 9 (PPV 100%). On the other hand, all the patients were HCV non-viremic if their anti-HCV titer was less than 9 (NPV 100%).

**Funding:** The authors received no specific funding for this work.

**Competing interests:** The authors have declared that no competing interests exist.

## Conclusions

Anti-HCV titer strongly predicted HCV viremia. This excellent performance could be generalized to either HCV mono-infected or HBV dually infected patients.

## Introduction

It is estimated that 71 million people are infected with hepatitis C virus (HCV) globally, which is a main public threat and disease burden. HCV is endemic in Taiwan with the seroprevalence of 2–4% [1,2]. From the viewpoint of natural history, 70%-80% of the subjects with HCV acquisition became chronically infected, 20% of whom progress to cirrhosis within 30 years. Among these cirrhotic patients, 25% of them would develop hepatocellular carcinoma (HCC) eventually. On the other hand, 20–30% of infected subjects spontaneously clear the virus within 6 months after acute infection [3–5].

Hepatic viral infections hepatitis B virus (HBV) is also rampant in Taiwan, which is the leading etiology of liver disease in Taiwan. Patients with HBV and HCV dual infection have a more deteriorated clinical course of liver disease progression, and are at an increased risk of HCC. Notably, both viruses may interact and suppress reciprocally, making the serological and virological diagnosis more complex [3,6]. The diagnosis of HCV infection usually starts from the screening test for HCV antibodies (anti-HCV) by methods of enzyme-linked immunosorbent assay (ELISA) and chemiluminescence immunoassay (CIA), and further confirmed by HCV RNA qualitative or quantitative testing. Detection of HCV-RNA by polymerase chain reaction (PCR) is considered the gold standard to confirm active HCV infection. Anti-HCV titer with different cut-off values has been proposed to predict HCV viremia [7,8]. Notably, whether the presence of HBV dual-infection affects the performance has never been studied. This study aimed to determine the optimal anti-HCV titer (S/CO ratio) in predicting HCV viremia in anti-HCV seropositive subjects with or without HBV dual-infection, and also sought to explore the accuracy of the anti-HCV titer with different cut-off values between patients with HCV mono-infected and dually infected patients.

## Methods

Eligible subjects were consecutively enrolled in the clinics of a medical center in Taiwan from 2013 to 2019. All the patients were tested for anti-HCV antibody, with the reason for testing anti-HCV being at the physicians' discretion. Patients were excluded if they had a history of antiviral therapy. The Institutional Review Board of the Kaohsiung Medical University Hospital approved the protocols that followed the guidelines of the International Conference on Harmonization for Good Clinical Practice (KMUHIRB-E(I)-20190301). All patients provided written informed consent.

Biochemical analyses were measured on a multichannel autoanalyzer (Hitachi Inc, Tokyo, Japan), while anti-HCV antibodies were detected using a commercially chemiluminescent microparticle immunoassay (Abbott Architect i-1000 system, Abbott Diagnostics, Lake Forest, Illinois, USA). Hepatitis B surface antigen (HBsAg) was determined using a standard quantitative chemiluminescent microparticle immunoassay (ARCHITECT HBsAg, Abbott Diagnostics). Patients who were seropositive for HBsAg and Anti-HCV were further tested for HBV DNA and HCV RNA respectively; then, HCV genotyping was further tested in HCV-viremic patients.

Quantitative serum HCV RNA levels and genotyping were measured using a standardized automated qualitative reverse transcription-polymerase chain reaction (RT-PCR) assay with a lower detection limit of 30 IU/mL and genotype was tested by HCV GT II. (Abbott Real-Time HCV Assay in m2000 PCR System) [9] HBV DNA viral load was detected by the COBAS® AmpliPrep/COBAS® TaqMan® HBV Test, v2.0, permitted automated specimen preparation followed by automated PCR amplification and detection of HBV target DNA with a lower detection limit of 20 IU/mL [10]. HCV mono-infection was defined as seropositive for anti-HCV. HBV dual-infection was defined as seropositive for both anti-HCV and HBsAg.

## Statistical analysis

Frequency was compared between groups using the $\chi^2$ test with the Yates correction or Fisher's exact test. Group means (presented as the mean standard deviation) were compared using analysis of variance and Student's *t*-test or the nonparametric Mann-Whitney test when appropriate. ROC curve was constructed by plotting sensitivity versus 1-specificity, using HCV-RNA testing and the S/CO ratio of anti-HCV respectively. AUROC was used to analyze the best cut-off value of anti-HCV titer in predicting HCV viremia, while anti-HCV titer with different cut-off values was also adopted to evaluate their accuracy in predicting HCV viremia. Statistical analyses were performed using IBM SPSS Statistics version 20 (IBM Corp., Armonk, NY, USA). All statistical analyses were based on two-sided hypothesis tests with statistical significance set at P<0.05.

## Result

### Patients

A total of 1321 anti-HCV seropositive patients were enrolled in the current study. Among them, 1049 (79.3%) patients were HCV RNA-positive. The mean age was 62.5 years, and males accounted for 45.6% of the population. Among the HCV-viremic patients, the mean HCV RNA was 5.6 log IU/mL and the most common HCV genotype was genotype 1 (43.8%) followed by HCV genotype 2 (30.4%). One hundred and twenty-five (9.5%) were HBV dual-infected. Compared to HCV mono-infected patients, those with HBV dual-infection had a lower proportion of HCV viremia (62.4% vs. 81.2%, P<0.001) and anti-HCV titer (10.1 S/CO vs. 12.0 S/CO). Of the 125 HBV dual infected patients, only 3 (2.4%) underwent nucleoside/nucleotide analogues (Table 1). Among the 1196 HCV viremic patients, the HCV RNA levels were 3.0 log IU/mL, 5.4 log IU/mL, 5.6 log IU/mL and 5.6 log IU/mL in subjects whose anti-HCV antibody titer were < 5 S/CO, 5–10 S/CO, 10–15 S/CO and > 15 S/CO, respectively (trend P = 0.13).

**Accuracy of anti-HCV titer in predicting HCV viremia.** The distribution of patients with different anti-HCV titers is shown in Table 2.

Among the entire population, the best cut-off value of anti-HCV titer in predicting HCV viremia was 9.95 (AUROC 0.99, P<0.0001) (S1A Fig). We further displayed the diagnostic performance of anti-HCV titer in predicting HCV viremia with different cut-off values. The different cut-off values of anti-HCV titer with the range from 5 to 15 could distinguish patients with or without HCV viremia significantly. The cut-off value of 9 or 10 provided the best accuracy, 96.8%. The sensitivity, specificity, positive predictive value (PPV) and negative predictive value (NPV) was 96.3%, 98.9%, 99.7% and 87.3% respectively, with the anti-HCV cut-off value of 10 in predicting HCV viremia (Table 3).

We further analyzed the characteristics of the patients with the extreme mismatch between anti-HCV titer and viremic status. As shown in the S1 Table, patients (n = 3) with anti-HCV S/CO <5 but HCVRNA (+) were older (68.0 years vs. 59.1 years, P = 0.0009) than the

**Table 1. Patient characteristics.**

| | All patients N = 1321 | HCV mono-infection, N = 1196 | HBV dual infection[†], N = 125 | P value |
|---|---|---|---|---|
| Age, years, mean (SD) | 62.5 (12.2) | 62.7 (12.2) | 62.7 (12.3) | 0.98 |
| Male gender, n (%) | 604 (45.7) | 538 (45.0) | 66 (52.8) | 0.10 |
| HCV RNA (+), n (%) | 1049 (79.3) | 971 (81.2) | 78 (62.4) | <0.0001* |
| HCV genotype, n (%) | | | | |
| 1 | 22 (2.1) | 22 (2.3) | - | |
| 1a | 55 (5.2) | 50 (5.1) | 5 (6.4) | |
| 1b | 501 (47.8) | 461 (47.5) | 40 (51.3) | |
| 2 | 401 (38.2) | 373 (38.4) | 28 (35.9) | |
| 3 | 14 (1.3) | 13 (1.3) | 1 (1.3) | |
| 4 | 1 (0.1) | 1 (0.1) | - | |
| 6 | 14 (1.3) | 13 (1.3) | 1 (1.3) | |
| Mixed | 38 (3.6) | 35 (3.6) | 3 (3.8) | |
| Unclassified | 3 (0.3) | 3 (0.3) | - | |
| AST, IU/L (median, range) * | 50 (11.2–1258) | 51 (11.2–1258) | 43 (19–404) | 0.98 |
| ALT, IU/L (median, range)* | 59 (7.4–1218) | 60 (7.4–1218) | 49 (14–544) | 0.64 |
| GGT, IU/L (median, range)* | 40 (8–1311) | 41 (8–1311) | 28 (8–454) | 0.28 |
| HCV RNA log IU/mL (mean, SD)^ | 5.6 (0.99) | 5.6 (0.99) | 5.5 (1.01) | 0.42 |
| HBV DNA log IU/ml (mean, SD) | 3.7 (1.7) | - | 3.7 (1.7) | - |
| Detectable HBV DNA, n (%)[#] | 49 (3.7%) | - | 49 (39.2%) | - |
| Anti-HCV titer in all patients, mean (SD) | 11.8 (4.7) | 12.0 (4.6) | 10.1 (5.5) | <0.001 |
| Anti-HCV titer in HCV viremic patients, mean (SD) | 13.9 (2.1) | 13.9 (2.1) | 14.1 (1.8) | 0.46 |

Note: HCV, hepatitis C virus; HBV, hepatitis B virus; AST, aspartate aminotransferase; ALT; alanine aminotransferase; GGT, gamma-glutamyl transferase; S/CO, signal-to-cut-off ratio

* AST and ALT data available in 1310 patients. GGT data available in 835 patients.

[†] 3 patients underwent nucleoside/nucleotide analogues.

^ Among HCV RNA seropositive patients.

\# lower detection limit: 20 IU/mL.

counterpart patients. By contrast, patients (n = 3) with anti-HCV >10 S/CO but HCVRNA (-) had lower levels of aspartate aminotransferase (23 IU/L vs. 59 IU/L, P<0.001), alanine amino-transferase (16 IU/L vs. 70 IU/L, P<0.001) and gamma-glutamyl transferase (16 IU/L vs. 42 IU/L, P<0.001) compared to the counterpart patients (S2 Table).

Among the HCV mono-infected patients, the best cut-off value of anti-HCV titer in pre-dicting HCV viremia was 9.95 (AUROC 0.99, P<0.0001) (S1B Fig). The best accuracy was 96.7% by using the cut-off value of 10, with the sensitivity, specificity, PPV and NPV of 96.2%, 98.7%, 99.7% and 85.7% respectively (Table 4).

Among the HBV dually infected patients, the best cut-off value of anti-HCV titer in predict-ing HCV viremia was 9.36 (AUROC 1.00, P<0.0001) (S1C Fig). The best accuracy was 100% by using the cut-off value of 9, with all the sensitivity, specificity, PPV and NPV reaching up to 100% respectively (Table 5).

## Discussion

In the current study, we demonstrated that anti-HCV titer provided excellent diagnostic per-formance for HCV viremia in Taiwanese patients. Anti-HCV titer with the cut-off value of 10 in HCV mono-infected and 9 in HBV dually infected patients provided the best accuracy. Notably, the role of anti-HCV titer in predicting HCV viremia seemed to exert more clinical

**Table 2. Anti-HCV titer of the patients.**

| S/CO, n (%) | All patients, n (%) N = 1321 | HCV mono-infection N = 1196 | HBV dual infection, N = 125 |
|---|---|---|---|
| 1–5 | 202 (15.3) | 167 (14.0) | 35 (28.0) |
| 5–6 | 19 (1.4) | 16 (1.3) | 3 (2.4) |
| 6–7 | 15 (1.1) | 10 (0.8) | 5 (4.0) |
| 7–8 | 25 (1.9) | 23 (1.9) | 2 (1.6) |
| 8–9 | 23 (1.7) | 21 (1.8) | 2 (1.6) |
| 9–10 | 24 (1.8) | 22 (1.8) | 2 (1.6) |
| 10–11 | 39 (3.0) | 35 (2.9) | 4 (3.2) |
| 11–12 | 65 (4.9) | 60 (5.0) | 5 (4.0) |
| 12–13 | 127 (9.6) | 118 (9.9) | 9 (7.2) |
| 13–14 | 195 (14.8) | 187 (15.6) | 8 (6.4) |
| 14–15 | 279 (21.1) | 255 (21.3) | 24 (19.2) |
| 15–16 | 196 (14.8) | 175 (14.6) | 21 (16.8) |
| 16–17 | 68 (5.1) | 65 (5.4) | 3 (2.4) |
| 17–18 | 31 (2.3) | 29 (2.4) | 2 (1.6) |
| 18–19 | 10 (0.8) | 10 (0.8) | 0 |
| 19–20 | 2 (0.2) | 2 (0.2) | 0 |
| 20–21 | 1 (0.1) | 1 (0.1) | 0 |

Note: S/CO, signal-to-cut-off ratio.

**Table 3. Diagnostic performance of Anti-HCV titer in predicting HCV viremia in the entire population.**

| | HCV RNA(-), N = 272(%) | HCV RNA(+), N = 1049(%) | P value | SEN(%) | SPE(%) | PPV(%) | NPV(%) | Accuracy(%) |
|---|---|---|---|---|---|---|---|---|
| ≧ 5 | 73 (26.8) | 1046 (99.7) | <0.001* | 99.7% | 73.2% | 93.5% | 98.5% | 94.3% |
| ≧ 8 | 27 (9.9) | 1033 (98.5) | <0.001* | 98.5% | 90.1% | 97.5% | 93.9% | 96.7% |
| ≧ 9 | 15 (5.5) | 1022 (97.4) | <0.001* | 97.4% | 94.5% | 98.6% | 90.5% | 96.8% |
| ≧ 10 | 3 (1.1) | 1010 (96.3) | <0.001* | 96.3% | 98.9% | 99.7% | 87.3% | 96.8% |
| ≧ 12 | 2 (0.7) | 907 (86.5) | <0.001* | 86.5% | 99.3% | 99.8% | 65.5% | 89.1% |
| ≧ 13 | 2 (0.7) | 780 (74.4) | <0.001* | 74.4% | 99.3% | 99.7% | 50.1% | 79.5% |
| ≧ 15 | 1 (0.4) | 307 (29.3) | <0.001* | 29.3% | 99.6% | 99.7% | 26.8% | 43.8% |

Note: HCV, hepatitis C virus; SEN, sensitivity; SPE, specificity; PPV, positive predictive value, NPV, negative predictive value.

**Table 4. Diagnostic performance of Anti-HCV titer in predicting HCV viremia in HCV mono-infected patients.**

| | HCV RNA(-), N = 225(%) | HCV RNA(+), N = 971(%) | P value | SEN(%) | SPE(%) | PPV(%) | NPV(%) | Accuracy(%) |
|---|---|---|---|---|---|---|---|---|
| ≧ 5 | 61 (27.1) | 968 (99.7) | <0.001* | 99.7% | 72.9% | 94.1% | 98.2% | 94.7% |
| ≧ 8 | 25 (11.1) | 955 (98.4) | <0.001* | 98.4% | 88.9% | 97.5% | 92.6% | 96.6% |
| ≧ 9 | 15(6.7) | 944 (97.2) | <0.001* | 97.2% | 93.3% | 98.4% | 88.6% | 96.5% |
| ≧ 10 | 3 (1.3) | 934 (96.2) | <0.001* | 96.2% | 98.7% | 99.7% | 85.7% | 96.7% |
| ≧ 12 | 2 (0.9) | 840 (86.5) | <0.001* | 86.5% | 99.1% | 99.8% | 63.0% | 88.9% |
| ≧ 13 | 2 (0.9) | 722 (74.4) | <0.001* | 74.4% | 99.1% | 99.7% | 47.3% | 79.0% |
| ≧ 15 | 1 (0.4) | 281 (28.9) | <0.001* | 28.9% | 99.6% | 98.7% | 24.5% | 42.2% |

Note: HCV, hepatitis C virus; SEN, sensitivity; SPE, specificity; PPV, positive predictive value, NPV, negative predictive value.

**Table 5. Diagnostic performance of Anti-HCV titer in predicting HCV viremia in HBV dual-infection patients.**

|  | HCV RNA(-), N = 47(%) | HCV RNA(+), N = 78(%) | value | SEN(%) | SPE(%) | PPV(%) | NPV(%) | Accuracy(%) |
|---|---|---|---|---|---|---|---|---|
| ≧ 5 | 12 (25.5) | 78 (100) | <0.001* | 100.0% | 74.5% | 86.7% | 100.0% | 90.4% |
| ≧ 8 | 2 (4.3) | 78 (100) | <0.001* | 100.0% | 95.7% | 97.5% | 100.0% | 98.4% |
| ≧ 9 | 0 (0) | 78 (100) | <0.001* | 100.0% | 100.0% | 100.0% | 100.0% | 100.0% |
| ≧ 10 | 0 (0) | 76 (97.4) | <0.001* | 97.4% | 100.0% | 100.0% | 95.9% | 98.4% |
| ≧ 12 | 0 (0) | 67 (85.9) | <0.001* | 85.9% | 100.0% | 100.0% | 81.0% | 91.2% |
| ≧ 13 | 0 (0) | 58 (74.4) | <0.001* | 74.4% | 100.0% | 100.0% | 70.2% | 84.0% |
| ≧ 15 | 0 (0) | 26 (33.3) | <0.001* | 33.3% | 100.0% | 100.0% | 47.5% | 58.4% |

Note: HCV, hepatitis C virus; SEN, sensitivity; SPE, specificity; PPV, positive predictive value, NPV, negative predictive value.

utility in HBV dually infected patients. Among them, all the patients were HCV-viremic if their anti-HCV titer was greater than 9. On the other hand, all the patients were HCV non-viremic if their anti-HCV titer was less than 9, indicating the potential surrogate of anti-HCV for active HCV infection in the special population.

Though we are in the fortunate era of directly-acting antivirals for HCV treatment [11–13], there existed tremendous gaps for HCV elimination [14,15]. Accurate and efficient HCV diagnosis possesses one of the hurdles of HCV care cascade [16,17]. Anti-HCV antibody is produced by antigen stimulation secondary to viral replication, and anti-HCV antibody level appears to be increased while the viral stimulation is persistent. Accordingly, the anti-HCV S/CO ratio is likely to be higher in patients with HCV viremia, in whom viral stimulation is strong and continuous, than that of patients with a history of past infection or with a waning infection [18].

Different anti-HCV S/CO ratio cut-off values ranging from 2.7 to 34 have been reported to determine HCV viremia by using the third generation of anti-HCV assays [7,8,19,20]. The difference cut-off values may in part attribute to different anti-HCV testing methods (ex. chemiluminescent microparticle immunoassay, microparticle enzyme immunoassay or enzyme-linked immunosorbent assay) (S3 Table). The current study was in line with a Korean study, which showed that anti-HCV S/CO ratio ~10 offered the best predictive power [8]. Both studies used the same chemiluminescent microparticle immunoassay. Despite its high accuracy in predicting HCV viremia, the test could not replace HCV RNA or core antigen test unless the virology testing is unavailable. Notably, whether the performance would be interfered by the presence of HBV or not, is unclear, as the two viruses might act interactively in the same host. We have previously demonstrated that the coexistence of HBV would enhance the chance of spontaneous HCV clearance as with the current study [3]. By contrast, HCV viremia may have a suppressive effect on both HBsAg and HBV DNA levels [6]. HCV often overwhelms HBV in the same host during the natural course [21]. The mean HBV DNA level was quite low in the present survey among HBV dually infected subjects, which was similar to our previous report [6]. The rate of detectable HBV DNA was around 40% in the current study, which was also not far from another community-based [3] or hospital-based [22] studies regarding HBV/HCV dual infection (<50%).

The correlation of anti-HCV titer and HCV viremia in patients with HBV dual-infection is elusive. In the current study, we demonstrated that the performance and cut-off value of anti-HCV in predicting HCV viremia was quite similar between HCV mono-infected and HBV dually infected patients. Intriguingly, among the HBV dual-infection patients, 100% of PPV and NPV could be provided with anti-HCV cut-off value of 9. It opens the room for discussion as to whether it could replace further HCV virology testing in this subpopulation in particular for patients residing in resource-restrained areas. Further study with larger patient numbers and heterogeneous patient characteristics could be warranted to validate the current findings.

We only tested HCV RNA once for patients who were seropositive for anti-HCV. We did not recheck HCV RNA at least 6 months apart thereafter. The current study was limited in failing to identify and exclude patients with acute hepatitis C infection in whom anti-HCV titer might be low despite the presence of HCV viremia. However, all the blood samples were retrieved from outpatient clinics during daily practice where acute hepatitis was less likely to be encountered. Albeit the case number should be rare, we also failed to identify subjects with occult HBV infection among HBsAg seronegative patients. In conclusion, anti-HCV titer strongly correlated to HCV viremia. The accurate performance could be generalized to either HCV mono-infected or HBV dual-infected patients.

## Supporting information

**S1 Fig. AUROC of the anti-HCV titer in predicting HCV RNA seropositivity.** A. All patients. B. HCV mono-infected patients. C. HBV dually infected patients.
(PDF)

**S1 Table. Comparison of patients whose anti-HCV S/CO <5 but HCVRNA (+) and their counterpart patients.**
(DOCX)

**S2 Table. Comparison of patients whose anti-HCV >10 S/CO but HCVRNA (-) and their counterpart patients.**
(DOCX)

**S3 Table. Studies of different anti-HCV testing methods and their cut-off values in predicting HCV viremia.**
(DOCX)

## Author Contributions

**Conceptualization:** Yi-Hung Lin.

**Data curation:** Ming-Lun Yeh, Po-Cheng Liang.

**Formal analysis:** Pei-Chien Tsai.

**Funding acquisition:** Ching-I Huang.

**Investigation:** Pei-Ju Lin, Zu-Yau Lin, Chia-Yen Dai.

**Methodology:** Shu-Fen Liu.

**Resources:** Ying-Chou Huang, Jee-Fu Huang, Ming-Lung Yu.

**Supervision:** Wan-Long Chuang.

**Visualization:** Jia-Jiun Tsai.

**Writing – original draft:** Hung-Yin Liu.

**Writing – review & editing:** Chung-Feng Huang.

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
