## [Decision Letter · Decision Letter 0]

17 May 2021

PONE-D-21-14163

Anti-HCV antibody titer highly predicts HCV viremia in patients with hepatitis B virus dual-infection

PLOS ONE

Dear Dr. Hung Yin Liu,

Thank you for submitting your manuscript to PLOS ONE. After careful consideration, we feel that it has merit but does not fully meet PLOS ONE’s publication criteria as it currently stands. Therefore, we invite you to submit a revised version of the manuscript that addresses the points raised during the review process.

We look forward to receiving your revised manuscript.

Kind regards,

Tatsuo Kanda, M.D., Ph.D.

Academic Editor

PLOS ONE

Journal Requirements:

2)  Thank you for stating the following financial disclosure:

 [The funders had no role in study design, data collection and analysis, decision to

publish, or preparation of the manuscript].

Reviewers' comments:

Reviewer's Responses to Questions

**Comments to the Author**

1. Is the manuscript technically sound, and do the data support the conclusions?

Reviewer #1: Partly

Reviewer #2: Partly

2. Has the statistical analysis been performed appropriately and rigorously? 

Reviewer #1: Yes

Reviewer #2: I Don't Know

3. Have the authors made all data underlying the findings in their manuscript fully available?

Reviewer #1: Yes

Reviewer #2: Yes

4. Is the manuscript presented in an intelligible fashion and written in standard English?

Reviewer #1: Yes

Reviewer #2: Yes

5. Review Comments to the Author

Reviewer #1: 1. Please mention if patients had history of anti-HBV therapy including interferon therapy.

2. As for patients with dual-infection, did all have HBV DNA positive? Please mention if there were any patient who had HBV DNA negative. Is there difference in HCV viral load between mono-infected and dual-infected patients?

3. Please mention whether all patients with HCV viremia were confirmed to have persistent infection.

4. Please provide HCV RNA viral load according to the anti-HCV antibody titer.

5. Please mention to what extent the different method of anti-HCV measurement affects the anti-HCV titer and cut-off value according to the previous reports.

6. In the discussion, please elaborate factors affecting low anti-HCV titer with HCV viremia or high anti-HCV titer with HCV negativity.

7. The method of HCV genotyping is not described.

Reviewer #2: In the manuscript titled “ Anti-HCV antibody titer highly predicts HCV viremia in patients with hepatitis B virus dual-infection”, the authors showed the valuable titer to predict Hepatitis C viremia in patiets with HCV mono-infection.

The authors defined HCV infection as HCV-Ab positive. I think that the definition of patients with HBV infection should include not only HBsAg positive but also HBsAb or HBcAb positive. There are patients whose HBsAg are negative but HBV-DNA are positive. HBsAg negative cannot exclude HBV infection.

6. PLOS authors have the option to publish the peer review history of their article (what does this mean?). If published, this will include your full peer review and any attached files.

Reviewer #1: No

Reviewer #2: No

---

## [Author Response · Author response to Decision Letter 0]

24 May 2021

Reviewer #1: 

1. Please mention if patients had history of anti-HBV therapy including interferon therapy.

Reply: Of the 125 HBV dually infected patients, only 3 (2.4 %) underwent nucleoside/nucleotide analogues. Since the case number was limited, it shall not have impact on the result regarding the issue. We have added it in the result of the revised manuscript. Thank you.

2. As for patients with dual-infection, did all have HBV DNA positive? Please mention if there were any patient who had HBV DNA negative. Is there difference in HCV viral load between mono-infected and dual-infected patients?

Reply: Thank you for the comment. Of the 125 HBV dually infected patients, forty-nine (39.2 %) had detectable HBV DNA (lower detection limit: 20 IU/mL). The HCV RNA level did not differ between patients with HCV monoinfection and HBV dual infection (5.6 log IU/mL vs. 5.5 log IU/mL, P=0.42). We have denoted it in the revised table 1 accordingly. 

3. Please mention whether all patients with HCV viremia were confirmed to have persistent infection.

Reply: Thank you for the great comment. The study was executed in the outpatient departments of the medical center. We only tested HCV RNA once for patients who were seropositive for anti-HCV. Most of the patients who were confirmed HCV viremia would be referred and allocated to antiviral therapy. We did not leave the viremic patients being observed and recheck HCV RNA at least 6 months apart thereafter. As mentioned in the text, we may fail to identify and exclude patients with acute hepatitis C infection. However, all the blood samples were retrieved from outpatient clinics during daily practice where acute hepatitis was less likely to be encountered. We have denoted it as the potential limitation in the current study. Thank you again for the comment. 

4. Please provide HCV RNA viral load according to the anti-HCV antibody titer.

Reply: Among the 1196 HCV viremic patients, the HCV RNA levels were 3.0 log IU/mL, 5.4 log IU/mL, 5.6 log IU/mL and 5.6 log IU/mL in patients whose anti-HCV antibody titer were < 5 S/CO, 5-10 S/CO, 10-15 S/CO and > 15 S/CO, respectively (trend P=0.13). We have denoted it in the revised manuscript.

5. Please mention to what extent the different method of anti-HCV measurement affects the anti-HCV titer and cut-off value according to the previous reports.

Reply: Thank you very much for the insightful comment. We fully agree with the review that different anti-HCV testing methods (ex. ELISA, CMIA or MEIA) may result in different cut-off values and accuracy in predicting HCV viremia. We have summarized previous reports and the current study in the supplementary Table 3. Notably, the current study was in line with the report by Seo et al [1]. who used the same chemiluminescent microparticle immunoassay and resulted in a similar cut-off value and accuracy. 

6. In the discussion, please elaborate factors affecting low anti-HCV titer with HCV viremia or high anti-HCV titer with HCV negativity.

Reply: Thank you for the great comment. As shown in the original Table 3, patients with the extreme mismatch between anti-HCV titer and viremic status were rare. We further analyzed the characteristics of the patients. As shown in Supplementary Table 1, patients (n=3) with anti-HCV S/CO <5 but HCVRNA (+) were older than the counterpart patients. By contrast, patients (n=3) with anti-HCV >10 S/CO but HCVRNA (-) had lower levels of aspartate aminotransferase, alanine aminotransferase and gamma-glutamyl transferase compared to the counterpart patients (supplementary Table 2). We have denoted in the revised manuscript. Thank you again.

7. The method of HCV genotyping is not described.

Reply: HCV genotyping was measured using a standardized automated qualitative reverse transcription-polymerase chain reaction (RT-PCR) assay (Abbott Real-Time HCV Assay in m2000 PCR System). We have denoted it accordingly.

---

## [Decision Letter · Decision Letter 1]

10 Jun 2021

PONE-D-21-14163R1

Anti-HCV antibody titer highly predicts HCV viremia in patients with hepatitis B virus dual-infection

PLOS ONE

Dear Dr. Hung Yin Liu,

Thank you for submitting your manuscript to PLOS ONE. After careful consideration, we feel that it has merit but does not fully meet PLOS ONE’s publication criteria as it currently stands. Therefore, we invite you to submit a revised version of the manuscript that addresses the points raised during the review process.

We look forward to receiving your revised manuscript.

Kind regards,

Tatsuo Kanda, M.D., Ph.D.

Academic Editor

PLOS ONE

Journal Requirements:

Reviewers' comments:

Reviewer's Responses to Questions

**Comments to the Author**

1. If the authors have adequately addressed your comments raised in a previous round of review and you feel that this manuscript is now acceptable for publication, you may indicate that here to bypass the “Comments to the Author” section, enter your conflict of interest statement in the “Confidential to Editor” section, and submit your "Accept" recommendation.

Reviewer #1: (No Response)

Reviewer #2: All comments have been addressed

2. Is the manuscript technically sound, and do the data support the conclusions?

Reviewer #1: (No Response)

Reviewer #2: Yes

3. Has the statistical analysis been performed appropriately and rigorously? 

Reviewer #1: (No Response)

Reviewer #2: Yes

4. Have the authors made all data underlying the findings in their manuscript fully available?

Reviewer #1: (No Response)

Reviewer #2: Yes

5. Is the manuscript presented in an intelligible fashion and written in standard English?

Reviewer #1: (No Response)

Reviewer #2: Yes

6. Review Comments to the Author

Reviewer #1: According to table 1, most HBsAg positive patients did not have detectable HBV DNA level.

I wonder if such infection stage frequently occurs in the natural course of HBV infection. How about HBsAg level in these patients?

Reviewer #2: (No Response)

7. PLOS authors have the option to publish the peer review history of their article (what does this mean?). If published, this will include your full peer review and any attached files.

Reviewer #1: No

Reviewer #2: No

---

## [Author Response · Author response to Decision Letter 1]

16 Jun 2021

Reviewer #1: According to table 1, most HBsAg positive patients did not have detectable HBV DNA level. I wonder if such infection stage frequently occurs in the natural course of HBV infection. How about HBsAg level in these patients?

Reply: We regret that HBsAg level was not available in the retrospective study. As mentioned in the discussion, there may exist reciprocally suppressive effect of the two viruses, and HCV often overwhelms HBV in the same host during the natural course.1 The mean HBV DNA level was quite low in the present survey, which was similar to our previous report.2 The rate of detectable HBV DNA was around 40 % in the current study, which was also not far from another community-based3 or hospital-based4 studies regarding HBV/HCV dual infection (<50 %). Patients were consecutively enrolled in the current study and we believed that there would be minimal selection bias. We have denoted it in the 2nd revised version. Thank you again for the great comment. 

References 

1. Raimondo G, Brunetto MR, Pontisso P, et al. Longitudinal evaluation reveals a complex spectrum of virological profiles in hepatitis B virus/hepatitis C virus-coinfected patients. Hepatology 2006;43:100-7.

2. Huang CF, Dai CY, Lee JJ, et al. Hepatitis C viremia interferes serum hepatitis B virus surface antigen and DNA levels in hepatitis B uremics. Hepatol Int 2014;8:224-232.

3. Yu ML, Dai CY, Huang CF, et al. High hepatitis B virus surface antigen levels and favorable interleukin 28B genotype predict spontaneous hepatitis C virus clearance in uremic patients. J Hepatol 2014;60:253-9.

4. Yeh ML, Huang CF, Huang CI, et al. Hepatitis B-related outcomes following direct-acting antiviral therapy in Taiwanese patients with chronic HBV/HCV co-infection. J Hepatol 2020;73:62-71.

---

## [Decision Letter · Decision Letter 2]

18 Jun 2021

Anti-HCV antibody titer highly predicts HCV viremia in patients with hepatitis B virus dual-infection

PONE-D-21-14163R2

Dear Dr. Hung Yin Liu,

We’re pleased to inform you that your manuscript has been judged scientifically suitable for publication and will be formally accepted for publication once it meets all outstanding technical requirements.

Kind regards,

Tatsuo Kanda, M.D., Ph.D.

Academic Editor

PLOS ONE

Additional Editor Comments (optional):

Reviewers' comments:

Reviewer's Responses to Questions

**Comments to the Author**

1. If the authors have adequately addressed your comments raised in a previous round of review and you feel that this manuscript is now acceptable for publication, you may indicate that here to bypass the “Comments to the Author” section, enter your conflict of interest statement in the “Confidential to Editor” section, and submit your "Accept" recommendation.

Reviewer #1: (No Response)

2. Is the manuscript technically sound, and do the data support the conclusions?

Reviewer #1: (No Response)

3. Has the statistical analysis been performed appropriately and rigorously? 

Reviewer #1: (No Response)

4. Have the authors made all data underlying the findings in their manuscript fully available?

Reviewer #1: (No Response)

5. Is the manuscript presented in an intelligible fashion and written in standard English?

Reviewer #1: (No Response)

6. Review Comments to the Author

Reviewer #1: (No Response)

7. PLOS authors have the option to publish the peer review history of their article (what does this mean?). If published, this will include your full peer review and any attached files.

Reviewer #1: No

---

## [Editor Report · Acceptance letter]

22 Jun 2021

PONE-D-21-14163R2 

Anti-HCV antibody titer highly predicts HCV viremia in patients with hepatitis B virus dual-infection 

Dear Dr. Liu:

I'm pleased to inform you that your manuscript has been deemed suitable for publication in PLOS ONE. Congratulations! Your manuscript is now with our production department. 

Kind regards, 

on behalf of

Dr. Tatsuo Kanda 

Academic Editor

PLOS ONE